# The Use of Hemispherical Directional Reflectance to Evaluate the Interaction of Food Products with Radiation in the Solar Spectrum

**DOI:** 10.3390/foods11131974

**Published:** 2022-07-02

**Authors:** Bartosz Błoński, Sławomir Wilczyński, Magdalena Hartman-Petrycka, Łukasz Michalecki

**Affiliations:** 1Department of Basic Biomedical Science, Faculty of Pharmaceutical Sciences in Sosnowiec, Medical University of Silesia, Kasztanowa Street 3, 41-200 Sosnowiec, Poland; b.blonski@sum.edu.pl (B.B.); mhartman@sum.edu.pl (M.H.-P.); 2Kornel Gibiński University Clinical Centre, Medical University of Silesia, Ceglana Street 35, 40-514 Katowice, Poland; lukaszmichalecki@gmail.com

**Keywords:** food packaging, directional hemispherical reflectance, radiation, chocolate

## Abstract

Food product packaging should block light to protect nutrients, color and active ingredients in functional food from degradation. Currently, packages are not optimized in terms of the solar radiation impact on the products they contain. The aim of this study was to develop a method of quantifying the interaction of food products with solar radiation, which would enable the optimization and selection of packaging that would protect the product from the spectral range specifically absorbed by it. In order to determine the reflectance of chocolate, the total reflectance ratio was measured. For this purpose, a SOC 410 Solar DHR reflectometer from Surface Optics Corporation, San Diego, CA, USA was used. Directional reflectance was measured for seven discrete spectral ranges from 335 to 2500 nm, which correspond to the spectrum of solar radiation. The value of total reflectance for chocolate differed significantly in the studied spectral ranges. The highest reflectance ratio, averaged for all the tested chocolate, was recorded for the spectral range 700–1100 nm and the lowest for the 335–380 nm range. The total reflectance was significantly correlated with the cocoa content and the brightness of the chocolate. The proposed method of hemispheric directional reflectance enables the measurement of the total reflectance of food products. It can be used as a measure of exposure to radiation. Thus, it is possible to design a package that will protect the product from the spectral range that is most harmful for it.

## 1. Introduction

Packaging today plays an important role in preserving the quality of food products, especially functional food, during their storage, providing protection against environmental, chemical and physical factors. The package can play a fundamental role in preventing product destruction, but it is also provides a barrier to water, oxygen, carbon dioxide and other gases as well as flavorings. In functional food, the food product package should block light to protect the nutrients and product color from deterioration as well as preventing the degradation of active ingredients. In addition to providing passive protection, many packages now play an active role in maintaining product quality, helping to preserve the desired atmosphere around the product [1,2,3,4].

The most important factor influencing the shelf life of food products, including functional food, is sunlight. Sunlight, including ultraviolet, visible and infrared radiation, leads to various degradation and oxidation reactions. The degradation and oxidation of food causes a decrease in the content of nutrients and bioactive compounds, the formation of unpleasant odors and flavors, changes in the color of the food, and the formation of toxic substances. Food compounds are sensitive to different wavelengths of light. Understanding the effect that specific wavelengths of light have on food ingredients will enable the development of new food packaging materials that block the most harmful wavelengths of light to the photostability of certain food compounds. However, at present, product packages are not optimized with regard to the effects of solar radiation on the finished product. One reasons is that consumers want to visually evaluate the food product before buying, which involves the need to produce transparent packages, or as in the case with chocolate, partially transparent packages. Transparent packaging enables product evaluation, but at the same time results in the exposure to natural and artificial light and increases the risk of food product degradation [5]. It should also be noted that non-transparent packages, such as, for example, cardboard, are also not optimized in relation to the product they are intended to protect. Packages transparent to visible radiation can be designed to protect against a selected range of UV or IR radiation.

Numerous studies on the influence of light energy on the stability and sensory properties of various food products can be found in the literature. The mechanism of photodegradation of many active ingredients, including vitamins, antioxidants and others, under the influence of solar radiation is also known. Nevertheless, there are only a few studies describing how individual spectral ranges of light affect the stability of final food products and not on individual ingredients. Such knowledge would make it possible to design a package that could selectively protect the product from the specific spectral range of light that would cause the most product degradation [6,7,8,9].

Light radiation only interacts with matter (food product, functional food) while being absorbed by it. The biological and chemical effects of reflected or transmitted light are zero. Thus, an important element determining the photostability of a food product is to establish to what extent and in what range it absorbs light radiation [5,10,11].

Absorption of radiation by food chromophores causes its transition into an excited state and then returns the absorbed energy to the food product. Solar radiation is absorbed for narrow spectral ranges when the difference in energy for specific molecular orbitals is equal to the energy (wavelength) of the radiation [10]. As an example, the diene, triene and tetraene systems for double bonds absorb radiation with a wavelength corresponding to ultraviolet 210, 233, 268 and 315 nm [11]. The spectral ranges for which these products absorb light radiation are known only for a small group of ready-made food products, an there is no chocolate in this group. Determining the interaction of a food product with light radiation of a given spectral range would enable optimization of the food packages in this respect and the selection of packages that would protect the product to the greatest extent from the spectral range specifically absorbed by the food product. This approach could revolutionize the packaging industry.

The aim of this study is to develop a new method for assessing the impact of solar radiation on food products, including functional food, using chocolate as an example. Chocolate is a relatively unstable food that degrades quickly when exposed to sunlight. The health-promoting properties of chocolate should also be taken into account, which it owes to numerous active ingredients that are also photosensitive. At the same time, chocolate is increasingly used as a functional food by enriching its composition with numerous minerals and vitamins.

## 2. Materials and Methods

### 2.1. Chocolate

A selection of 14 different chocolates were tested (Table 1). Up-tempered and sucrose-free chocolates were also tested. Chocolate cubes with dimensions of at least 20 × 20 × 16 mm and weight 11.6–12.3 g were tested while analyzing the smooth surface. The products were selected to represent all types of chocolate (dark, milk, dessert), with a spectral range of cocoa content, as well as sucrose-free chocolate and chocolate enriched with minerals and vitamins.

### 2.2. Directional Hemispherical Reflectance 

In order to determine the reflectance of the tested chocolate, the total directional reflectance ratio was determined. For this purpose, a SOC 410 Solar DHR reflectometer from Surface Optics Corporation, San Diego, USA was used. The SOC 410 Solar equipped with a DHR (Directional Hemispherical Reflectance) measurement head measures the integrated surface reflectance of a surface at the angle of 60° and for seven discreet wavelength bands in the 335 nm to 2500 nm spectral range. The integrating sphere captures the reflected radiation from the target material (chocolate), integrating reflections in all directions. Wavelength filtered detectors measure the total radiation reflected in each wavelength band and convert it to an analog electrical signal. 

The total directional reflectance of a surface is defined as the ratio of the total energy reflected into the subtending hemisphere to the energy incident on the studied surface. 

Following the notation of Nicodemus [11], the directional reflectance may be expressed in terms of primary quantities as:$$\rho_{d}\left(\Theta_{i},\phi_{i}\right)=\frac{\int_{0}^{2\pi }\int_{0}^{\pi /2}N_{r}\sin\Theta_{r}\cos\Theta_{r}d\Theta_{r}d\phi_{r}}{N_{i}\sin\Theta_{i}\cos\Theta_{i}d\Theta_{i}d\phi_{i}}$$
where:

*ρ_d_* is the reflectance;

*Θ_i_, ϕ_i_* is the direction of energy incident on the surface (Figure 1); 

*N_i_* is a radiance function of both position and direction, incident on the surface of an opaque object where some of the radiation is absorbed and the rest is reflected (includes diffuse reflectance or scattering) (Figure 1); 

*N_r_* is the radiance of the reflected radiation (also a function of position and direction).

Directional reflectance was tested for seven discrete spectral ranges for the 14 tested chocolates. For each measurement, the chocolate surface was sampled three times on each side of the cube for seven spectral ranges: 335–380 nm, 400–540 nm, 480–600 nm, 590–720 nm, 700–1100 nm, 1000–1700 nm and 1700–2500 nm.

In order to determine the absolute reflectance for the tested chocolate, two calibration coupons certified by the ANIST (American National Institute of Standards and Technology) were used. The reflectance was determined according to these standards.

Light radiation can be reflected, scattered and/or absorbed by the surface of the chocolate. The quantitative evaluation of the scattering or reflection of light radiation from chocolate makes it possible to determine how much radiation is absorbed by the object and thus determine the strength of the impact of given spectral ranges on the food product. This will make it possible to optimize the radiation protection properties of packages in relation to the spectral properties of the food product.

### 2.3. Emissivity 

The obtained reflectance values can be used to determine the emissivity of the tested objects according to the relationship:ε = 1 − ρ
where:

ε is the object emissivity;

ρ is the object reflectance.

Emissivity is defined as the ratio of the energy radiated from a material’s surface to that radiated from a perfect emitter, known as a blackbody, at the same temperature and wavelength and under the same viewing conditions. It is a dimensionless number between 0 (for a perfect reflector) and 1 (for a perfect emitter). 

Thus, in order to accurately determine the heat transfer capacity between the environment and the surface of a given material (object), its emissivity should be identified. Because the reflectometer used in this study emits radiation, the emission of which (both in the wavelength and intensity for individual spectral ranges) is equivalent to solar radiation, one chocolate emissivity ratio was determined for solar radiation.

### 2.4. Brightness Analysis

Brightness analysis was performed using ImageJ software. ImageJ is Java-based open-source software developed at the National Institutes of Health and the Laboratory of Optics and Computational Instrumentation (LOCI, University of Wisconsin).

Before determining the level of brightness of the chocolate image, the image was captured using the Fotomedicus clinical photography system, Elfo, Poland. The main element of the system is a spherical cap with a Canon EOS 350D camera and a ring flash with constant flash energy (the flash energy is current-controlled, and the light used has a very high color rendering index of >90%). The Fotomedicus system enables the acquisition of images in non-polarised and cross-polarised light for the polarisation angle of θ = 90°. Such polarisation is achieved through the use of two linear polarising filters. The first filter is placed on the camera lens and the other one on the light sources. The images obtained in this way were saved in a lossless RAW format. The normalization of the recorded images was based on the identification of the brightest and darkest pixel in the entire set of recorded images. The lightest pixel was assigned the brightness of 1, and the darkest pixel was assigned the brightness of 0. This made it possible to reduce the changes in the brightness of pixels to the full range of gray levels (range from 0 to 255). In this way, the average brightness in each of the images was determined for each chocolate sample. At the same time, the level of gray in the obtained density standards was determined, which were recorded simultaneously with the tested samples. On this basis, the average brightness of the tested chocolate was identified.

### 2.5. Statistical Analysis

The following statistical tests were used: Shapiro–Wilk to assess the normality of the distribution; Friedman, ANOVA and Dunn’s post hoc to assess the reflectance differences in different wavelength ranges; Spearman’s correlation to assess the relations between cocoa content and chocolate brightness, between chocolate brightness and reflectance, and between cocoa content in chocolate and reflectance.

## 3. Results

The idea of optimizing the packages in relation to the spectral characteristics of the object (measured with the use of the total reflectance ratio) is as follows: for a food product (functional food) the most harmful is the spectral range absorbed by the object to the greatest extent. Only the absorbed radiation has a biological effect. Thus, the lower the reflectance ratio for a given spectral range of a food product, the more effective protection against this range is required; the package should eliminate this range of radiation.

The chocolate reflectance values differed significantly depending on the wavelength ranges (*p* ≤ 0.001, the results of the post hoc test are shown in Figure 2). The lowest values of chocolate reflectance occurred at the wavelengths of 335–380 nm (median 0.067), 400–540 nm (median 0.097), 480–600 nm (median 0.074) and 1700–2500 nm (median 0.080). The highest reflectance values occurred in the wavelength range of 700–1100 nm (median 0.284) and 1000–1700 nm (median 0.239). Intermediate reflectance values occurred at a wavelength of 590–720 nm (median 0.123). Due to the lack of a normal distribution, the standard deviation was not used as a measure of the spread, but the median and quartile range values were given, which are appropriate for this type of data. 

The cocoa content in the tested chocolate ranged from 30% to 85%, and the median was 60% (Figure 3, Table 2). The brightness of the chocolate ranged from 43.0 to 71.1 with a median value of 53.7. There is a significant statistical correlation between the cocoa content and the brightness of the chocolate (Figure 3). The lower the cocoa content, the brighter the color of the chocolate (R = −0.82, *p* ≤ 0.001).

The reflectance of chocolate is very much dependent on the brightness of the chocolate (Figure 4). It was shown that the brighter the chocolate, the higher the reflectance. This dependence was observed for almost all ranges of wavelength, except for the wavelength in the range of 1700–2500 nm. Taking into account the other wavelength ranges, the least statistically significant correlation of color and reflectance occurred at the wavelength of 335–380 nm (R = 0.66, *p* = 0.010). All wavelength ranges from 400 to 1700 nm showed a highly significant, very strong correlation between color and reflectance (R ≥ 0.80, *p* ≤ 0.001).

A higher cocoa content in the chocolate reduces the reflectance for some wavelength ranges (Figure 5). The strongest correlations of cocoa content and reflectance occurred at 400–540 nm (R = −0.56, *p* = 0.036), 700–1100 nm (R = −0.58, *p* = 0.031) and 1000–1700 nm (R = −0.58, *p* = 0.029). The correlation of cocoa content and reflectance at the wavelengths of 480–600 nm was close to the level of statistical significance, but did not reach it (R = −0.52, *p* = 0.054). At the wavelengths of 335–380 nm, 590–720 nm and 1700–2500 nm, there was no significant relation between the cocoa content and reflectance. 

The emissivity of chocolate, understood as the average emissivity for all tested chocolates in the studied spectral range (335–2500 nm) was 0.85.

## 4. Discussion

This new method applied to assess the optical properties of food products, including functional food, using chocolate as an example, enables the determination of the reflectance ratio in a wide spectral range. It is particularly important to determine the spectral properties of functional food in terms of solar radiation as this range is practically important in terms of the durability and usefulness of functional food.

The effects of sunlight-induced photooxidation in food can be broadly classified into four main areas: effects on food quality, effects on nutritional properties, effects on active ingredient content, and effects on health due to potential toxicity [12].

The tested chocolates consisted mainly of fats and carbohydrates. In general, the effect of light energy on food lipids is related to the formation of volatile compounds responsible for bad odors and tastes of food products. Food products high in polyunsaturated fatty acids are susceptible to oxidation by singlet oxygen. The degradation of carbohydrates under the influence of solar radiation in chocolate is also related to the action of singlet oxygen. Singlet oxygen reacts primarily with three essential amino acids (tryptophan, methionine and histidine) as well as cysteine and tyrosine. These amino acids, which contain double bonds or sulfur atoms in their structures, easily react with singlet oxygen that has electron affinity [13]. This makes chocolate particularly sensitive to solar radiation and it degrades very quickly under its influence.

The proposed method for the assessment of total reflectance for food products makes it possible to determine the amount of radiation absorbed or reflected by the product in a given spectral range, and thus enables the optimization of product packages. In the case of the tested chocolate, the highest reflectance was found for the spectral range of 700–1100 nm, amounting to about 28% (average for all chocolates). This means that about 70% of the radiation in this spectral range can interact with chocolate. On the other hand, for the spectral range of 335–380 nm, only less than 1% of the radiation is reflected or scattered by the tested chocolate. This means that protection against this particular wavelength should be a priority and packaging should especially block radiation in this spectral range.

However, it should also be taken into account that the radiation energy is inversely proportional to the radiation wavelength and thus the energy it carries. Therefore, chocolate will not only be the most exposed to degradation under the influence of ultraviolet radiation (the lowest reflectance ratio in the range of 335–380 nm), but at the same time the amount of energy carried by this radiation will be proportionally higher than, for example, for radiation in the UVA range. Thus, the radiation energy in the spectral range of 335–380 nm will affect chocolate 1250% more intensely (12.5 times more) than the radiation in the spectral range of 700–1100 nm.

The determination of total reflectance ratios for food products can also be used to optimize the lighting of shop windows and shelves where chocolate or other functional foods are displayed. It would be possible to use selective filters on lighting sources to optimize the lighting in a way that causes the least degradation of food products, including the tested chocolate.

The total reflectance ratio of all tested chocolate for individual spectral ranges is similar. However, there is a clear correlation between chocolate brightness, cocoa content and reflectance. The more cocoa or darker the chocolate, the better its protection should be because the total reflectance ratio is lower. This dependence was observed for almost all ranges of wavelength, except for the wavelengths in the range of 1700–2500 nm. It should be noted that this effect is likely related to the very physical effects occurring between radiation and chocolate. The longer the radiation length, the deeper it penetrates the object (assuming that the object does not contain chromophores absorbing a specific range of radiation). Importantly, the cocoa content, although it is a very convenient parameter for verification because it is declared by chocolate manufacturers, is not an unambiguous parameter. The concept of cocoa includes the sum of cocoa butter and cocoa liquor. The cocoa liquor gives the chocolate a dark color.

In the conducted tests, the total reflectance was determined for the chocolate samples by sampling the smooth part of the chocolate cube wall three times. A very small standard deviation (less than 5%) indicates that the spectral properties of the chocolate are the same in all directions, i.e., the homogeneity of the samples is high. In view of the above, the directional reflectance analysis may be considered in the future as a method for assessing the homogeneity of chocolate. Nevertheless, this requires other research that goes beyond this publication.

The functional food packages, including for chocolate, should also be optimized to protect active ingredients. Absorption of radiation in a given spectral range may reduce the content of vitamins and antioxidants. Flavonoids are important active ingredients of chocolates [14,15,16], and the most destructive spectral range for flavonoids is the wavelength of about 320–360 nm, where they are particularly vulnerable to photoxidation [17,18]. It is especially important because less than 7% of the radiation in this spectral range is reflected or scattered by the tested chocolate. Therefore, protection against this range of radiation should be as effective as possible. Nevertheless, one advantage of this situation is the fact that radiation of this wavelength has only a limited possibility of penetrating into objects with the density of chocolate.

The most effective way to protect food products from degradation is to block light access completely. Nevertheless, this approach limits the visibility of the product to the consumer. Therefore, the packages of food products should be designed in such a way as to protect the active ingredients as much as possible. The proposed method for assessing the total reflectance of food products, using chocolate as an example, makes it possible to quantify what spectral range is absorbed or reflected by the food product and thus to determine the degree of exposure to damage during storage. This method has many advantages: it is quantitative, reproducible, non-invasive (the sample is not degraded), fast and universal.

Understanding the effects of specific wavelengths of light on food ingredients in the food industry will enable the development of new, highly tailored, food packaging materials. The knowledge of the interaction with radiation of the active ingredients themselves is not sufficient; the interaction with solar radiation of the entire finished product must also be understood. Particular attention in the protection of food products, including the tested chocolate, should be given to those radiation ranges from the sunlight spectrum that are absorbed by the finished product to the greatest extent. At the same time, even relatively small changes in the packaging, for example the thickness of cardboard, can significantly affect its spectral properties.

Importantly, the aperture of the reflectometer used has a diameter of 7 mm. As a result, the measurement area is relatively large, and thus local aberrations in the composition of the product will not affect the result. Compared with the method used with classical spectroscopy, where the measuring spot usually has an area below 1 mm^2^, it is a significant advantage. This is due to the fact that the depth of penetration of the radiation depends not only on the wavelength of the radiation, as already noted, but also on the size of the spot. The larger the spot, the deeper the radiation penetrates into the object. At the same time, the proposed method has an advantage over spectrophotometric methods because in the spectrophotometer, the beam is directed at the object at an angle of 90°. In real conditions, the radiation does not fall on the object at this angle, hence the 20° angle in the reflectometer better mirrors the real conditions.

It should be emphasized that apart from the undoubted advantages indicated above, the proposed method has a number of limitations. First of all, the directional reflectometer is relatively expensive, which limits the possibility for its widespread use. At the same time, the research was carried out in a slightly narrower spectral range than solar radiation because the range of solar radiation covers the spectrum from 250 nm. In addition, the pre-selected spectral ranges of the reflectometer limit the possibility of optimizing the packaging in relation to specific products with a narrow range of radiation absorption. Another limitation is the necessity to conduct tests on a flat surface of the chocolate (without any embossing), and this surface must be larger than the diameter of the reflectometer aperture (7 mm).

## 5. Conclusions

The new method presented here of using directional hemispherical reflectance allows the measurement of total reflectance of food products and also functional food including chocolate. Assessing the spectral ranges that correspond to the maximum absorption of radiation by the food product can help optimize packaging selection. The packaging should protect against the spectral range of the product which is most strongly absorbed by the food product (it is subject to the lowest reflectance). In the case of chocolate, it is the spectral range 335–380 nm. The use of directional reflectance has a number of advantages over spectrophotometric methods for evaluating the spectral properties of food: It allows for the analysis of reflectance for samples that are not transparent in the studied radiation ranges; it interacts with the object in way closer to the actual conditions of incidence of solar radiation; and it has a larger spot (aperture), and thus more effectively reflects the real conditions. The proposed method may be an interesting starting point for attempts to correlate the spectral properties of products with the spectral properties of the packaging.

## Figures and Tables

**Figure 1 foods-11-01974-f001:**
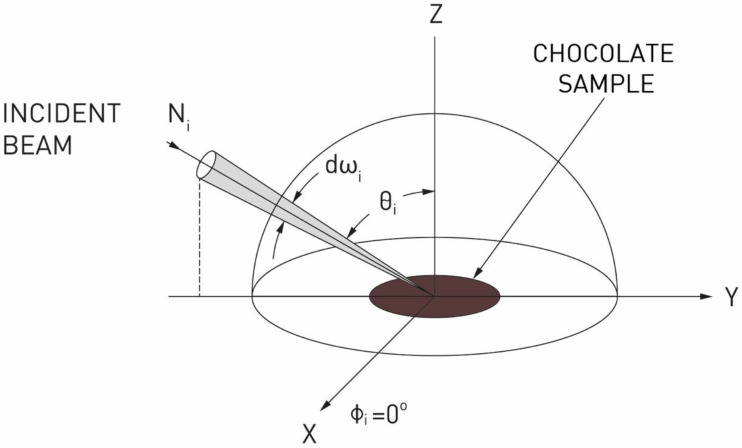
Diagram illustrating the concept of directional reflectance measurements of chocolate.

**Figure 2 foods-11-01974-f002:**
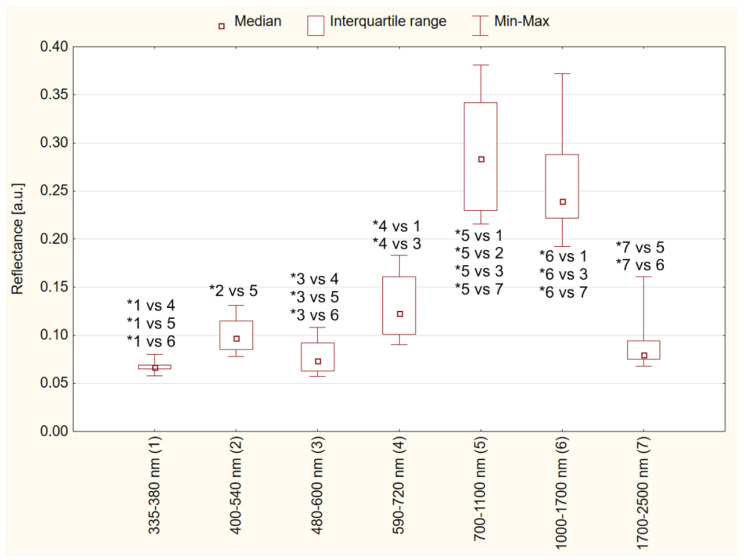
Chocolate reflectance values at different wavelength ranges; (1–7) wave range number * *p* ≤ 0.05.

**Figure 3 foods-11-01974-f003:**
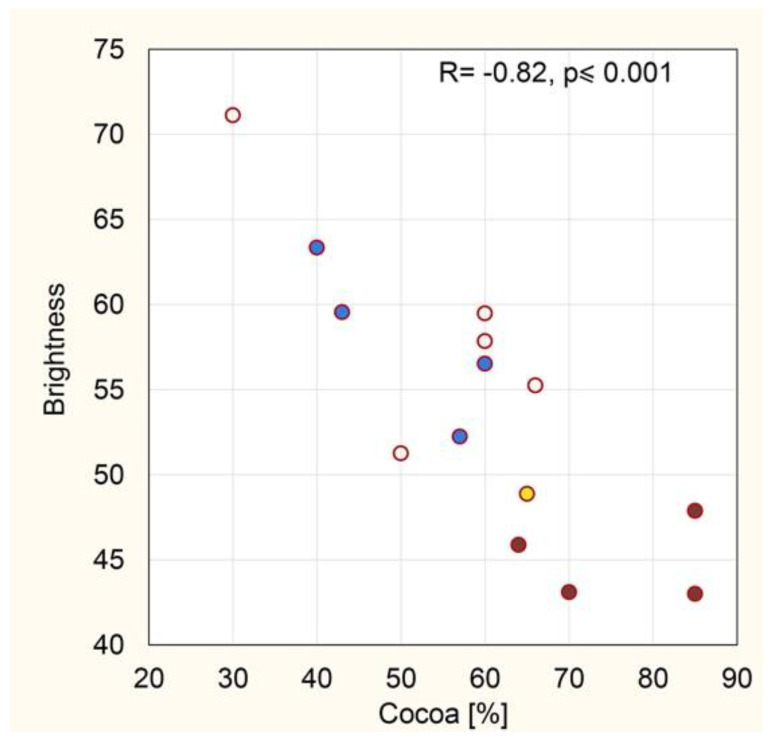
Correlation between the brightness of the chocolate and the cocoa content in the chocolate. Brown dark chocolate, blue milk chocolate, white dessert chocolate, yellow Viminella.

**Figure 4 foods-11-01974-f004:**
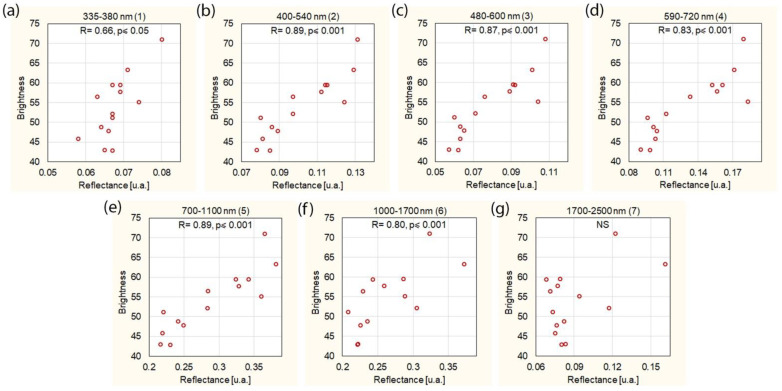
Correlation between chocolate brightness and reflectance for different wavelength ranges: (**a**) 335–380 nm, (**b**) 400–540 nm, (**c**) 480–600 nm, (**d**) 590–720 nm, (**e**) 700–1000 nm, (**f**) 1000–1700 nm, (**g**) 1700–2500 nm. NS—not significant.

**Figure 5 foods-11-01974-f005:**
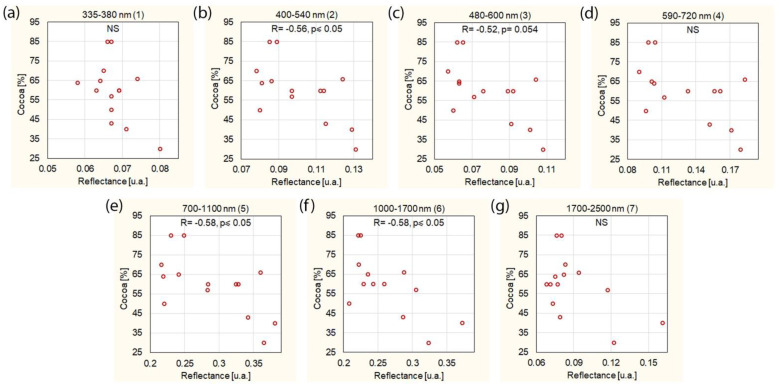
Correlation between cocoa content in chocolate and reflectance for different wavelength ranges: (**a**) 335–380 nm, (**b**) 400–540 nm, (**c**) 480–600 nm, (**d**) 590–720 nm, (**e**) 700–1000 nm, (**f**) 1000–1700 nm, (**g**) 1700–2500 nm. NS—not significant.

**Table 1 foods-11-01974-t001:** Tested chocolates with cocoa content declared by the manufacturer.

Sample Number	Chocolate Name	Cocoa Content
1	Milk, Wedel	30%
2	Milk with stevia, Bartfan	40%
3	Dessert, Wawel	43%
4	Dessert, Wedel	50%
5	Milk, Roshen	57%
6	Milk, Simon Coll	60%
7	Dark with fructose, Bartfan	60%
8	Dessert, Ghirardelli	60%
9	Dark, Wedel	64%
10	Viminella, Bartfan	65%
11	Dessert, Baker’s	66%
12	Dark with fructose, Bartfan	70%
13	Dark, un-tempered, Bartfan	85%
14	Dark, Lindt	85%

**Table 2 foods-11-01974-t002:** The brightness of the chocolate and the cocoa content in the chocolate.

	Median	1st Qartil	3rd Quartil	Minimum	Maximum
Brightness	53.7	47.9	59.5	43.0	71.1
Cocoa [%]	60	50	66	30	85

## Data Availability

The data presented in this study are available on request from the corresponding author.

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
