# Peer review of "The Use of Hemispherical Directional Reflectance to Evaluate the Interaction of Food Products with Radiation in the Solar Spectrum"

_foods, 2022, doi:10.3390/foods11131974_

Round 1

Reviewer 1 Report

The manuscript foods-1747230,  Protection of functional food against solar radiation - optimization of the packaging in relation to the spectral properties of functional food on the example of chocolate -  describes the developing of a method of quantifying the interaction of food product- chocolate - with solar radiation, method that will make possible the optimization and selection of the package that would better protect the product from the spectral range specifically absorbed by it.

The abstract of the paper is well written with a single observation - the entire solar spectrum is 250 nm to 2500 nm. (325-2500 nm is investigated solar spectrum)

The introduction is well-written and summarizes recent research related to the topic and highlights the importance of the topic and the main gaps in current understanding.

Line 29   Please reformulate the sentence ”Providing protection against environmental, chemical and physical factors.”!

Materials and methods.

Line 90 – “Chocolate cubes with dimensions of at least 20 x 20 mm were tested while analyzing the smooth surface.” If you are referring to chocolate cubes then it is enough to just say the length of the side, if they are chocolate cuboid then you will need another extra size to specify

Concerning Table 1. Tested chocolate with cocoa content declared by the manufacturer.- I did not identify the criteria by which the samples were presented in Table 1. It would be desirable to present them either by type of chocolate (milk, dark, dessert) or by increasing cocoa content.

Line 107 The quoted reference [11] has nothing to do with the context in which it is quoted. Please correct!

Line 112 It refers to a figure without any explanation (Fig.1)

Line 115 It also refers to a figure that not exist (Fig.1) This has probably been dropped. Please correct this!

Line 123 Please correct in the following way: “coupons certified by the ANIST (American National Institute of Standards and Technology)”

Line 125 Please reformulate “Light radiation can reflect/scatter and/or absorb this radiation on the surface of the chocolate.” It is not clear!

Line 152 I did not find any information about Fotomedicus clinical photography system, Elfo, Poland. Please provide more information, camera type, photosensitive element type ... etc!

Line 154 What means “a very high color rendering index.” How much is CRI? What kind of lamp is, incandescent or halogen?

In my opinion, the brightness measurement and evaluation system does not have a good enough scientific basis. There are elements that can significantly alter the measurements made, namely the spectral composition of the illuminator, the sensitivity and the calibration curve of the receiver / camera. Please scientifically argue how you measure brightness, otherwise it looks like a craft and not a scientific method. What is the spectrum of the illuminating lamp used?

Results

Figure 1. Replace the comma in the labels on the ordinate with the decimal point!

In Figure 2 the graphs are not marked with abscissas and ordinates and are very difficult to follow them. Also there are no units. Please correct this!

Why did you choose to make a global brightness assessment based only on the cocoa content and not on the type of chocolate milk, dark or dessert? For example, a 60% cocoa content because the measurements were made on three types of chocolate seems to led to the greatest dispersion of results.

 Figure 3 is very unclear ... it is not clear who the data sections represented in the form of bars correspond to or what correlations are made. Even if the authors, through an image analysis performed with the Image J software, quantify the brightness of the captured images, the authors come to discover a well-known physical truth, namely that brightness is reflectance measured in visible radiation, and that brightness correlates with reflectance to a very large extent. .

In conclusion, the paper has serious scientific problems, does not contribute to the development of a credible method for assessing the ability of packaging to protect food in this case chocolate from radiation and I do not recommend it for publication in a prestigious journal such as Foods.

Author Response

Dear Reviewer,

I am very grateful for the valuable opinions and remarks. I agree with all the comments and I have referred to them as best as possible in a revised version of the manuscript. List of corrections is presented below:

  • The manuscript foods-1747230,  Protection of functional food against solar radiation - optimization of the packaging in relation to the spectral properties of functional food on the example of chocolate -  describes the developing of a method of quantifying the interaction of food product- chocolate - with solar radiation, method that will make possible the optimization and selection of the package that would better protect the product from the spectral range specifically absorbed by it.

The abstract of the paper is well written with a single observation - the entire solar spectrum is 250 nm to 2500 nm. (325-2500 nm is investigated solar spectrum)

The introduction is well-written and summarizes recent research related to the topic and highlights the importance of the topic and the main gaps in current understanding.

Line 29  Please reformulate the sentence ”Providing protection against environmental, chemical and physical factors.”!

The sentence was reformulate to:

Packaging today plays an important role in preserving the quality of food products, especially functional food, during their storage, providing protection against environmental, chemical and physical factors.

  • Materials and methods.

Line 90 – “Chocolate cubes with dimensions of at least 20 x 20 mm were tested while analyzing the smooth surface.” If you are referring to chocolate cubes then it is enough to just say the length of the side, if they are chocolate cuboid then you will need another extra size to specify

The information has been added: Chocolate cubes with dimensions of at least 20 x 20 x 16 mm  and weight 11.6 – 12.3 g were tested while analyzing the smooth surface.

  • Concerning Table 1. Tested chocolate with cocoa content declared by the manufacturer.- I did not identify the criteria by which the samples were presented in Table 1. It would be desirable to present them either by type of chocolate (milk, dark, dessert) or by increasing cocoa content.

The table was corrected – chocolates were presented by increasing coca content.

Sample number

Chocolate name

Cocoa content

1

Milk, Wedel

30%

2

Milk with stevia, Bartfan

40%

3

Dessert, Wawel

43%

4

Dessert, Wedel

50%

5

Milk, Roshen

57%

6

Milk, Simon Coll

60%

7

Dark with fructose, Bartfan

60%

8

Dessert, Ghirardelli

60%

9

Dark, Wedel

64%

10

Viminella, Bartfan

65%

11

Dessert, Baker’s

66%

12

Dark with fructose, Bartfan

70%

13

Dark, un-tempered, Bartfan

85%

14

Dark, Lindt

85%

  • Line 107 The quoted reference [11] has nothing to do with the context in which it is quoted. Please correct!

Corrected to the appropriate reference: Nicodemus, F. Directional reflectance and emissivity of an opaque surface. Appl Opt 1965, 4(7), 767–773.

  • Line 112 It refers to a figure without any explanation (Fig.1)

An additional figure adequate to the description has been added.

Fig. 1 Diagram illustrating the concept of directional reflectance measurements of chocolate.

  • Line 115 It also refers to a figure that not exist (Fig.1) This has probably been dropped. Please correct this!

Corrected.

  • Line 123 Please correct in the following way: “coupons certified by the ANIST (American National Institute of Standards and Technology)”

Corrected.

  • Line 125 Please reformulate “Light radiation can reflect/scatter and/or absorb this radiation on the surface of the chocolate.” It is not clear!

Corrected to: Light radiation can be reflected / scattered and / or absorbed by the surface of the chocolate.

  • Line 152 I did not find any information about Fotomedicus clinical photography system, Elfo, Poland. Please provide more information, camera type, photosensitive element type ... etc!

The information has been added: The main element of the system is a spherical cap with a Canon EOS 350D camera and a ring flash with constant flash energy (the flash energy is current-controlled and the light used has a very high colour rendering index of >90%). The Fotomedicus system enables the acquisition of images in non-polarised and cross-polarised light for the polarisation angle of θ = 90°. Such polarisation is achieved through the use of two linear polarising filters. The first filter is placed on the camera lens and the other one on the light sources. The images obtained in this way were saved in a lossless RAW format.

  • Line 154 What means “a very high color rendering index.” How much is CRI? What kind of lamp is, incandescent or halogen?

In my opinion, the brightness measurement and evaluation system does not have a good enough scientific basis. There are elements that can significantly alter the measurements made, namely the spectral composition of the illuminator, the sensitivity and the calibration curve of the receiver / camera. Please scientifically argue how you measure brightness, otherwise it looks like a craft and not a scientific method. What is the spectrum of the illuminating lamp used?

Of course, we agree with the Reviewer's opinion that all the mentioned elements, such as spectral composition of the illuminator, the sensitivity and the calibration curve of the receiver / camera, are very important in assessing the brightness of objects. Thus, it is difficult to determine the optimal conditions for image acquisition in a repeatable manner for other laboratories, including, for example, the light incidence angle, peak lamp power, studio conditions (light scattering on other objects, which may interfere with the recorded image). Therefore, pictures of all samples were recorded under the same conditions - in one photo. Thanks to this, it was possible to avoid the problems described above.

  • Results

Figure 1. Replace the comma in the labels on the ordinate with the decimal point!

  • In Figure 2 the graphs are not marked with abscissas and ordinates and are very difficult to follow them. Also there are no units. Please correct this!

Corrected.

  • Why did you choose to make a global brightness assessment based only on the cocoa content and not on the type of chocolate milk, dark or dessert? For example, a 60% cocoa content because the measurements were made on three types of chocolate seems to led to the greatest dispersion of results.

In the adopted research model it was assumed that one variable (cocoa content) would be a more effective factor determining the color of chocolate. In addition, the cocao content was chosen due to the easier use of this parameter in the future. The content of other ingredients that may affect the color of the chocolate is not specified on the packaging and is often a trade secret of the company. Hence the concept of correlating color with cocao content. The authors realize that the cocao content is also not a perfect parameter, if only because cocao is understood as a mixture of the sum of cocoa butter (cocoa butter) and cocoa liquor. Nevertheless, it is a parameter that is of great practical importance - hence its choice.

  • Figure 3 is very unclear ... it is not clear who the data sections represented in the form of bars correspond to or what correlations are made. Even if the authors, through an image analysis performed with the Image J software, quantify the brightness of the captured images, the authors come to discover a well-known physical truth, namely that brightness is reflectance measured in visible radiation, and that brightness correlates with reflectance to a very large extent.

Of course, we agree with the reviewer's statement that brightness correlates with reflectance to a very large extent. However, other parameters - including the material - also affect the reflectance. The attempt to correlate the brightness of chocolate with reflectance was aimed at assessing the possibility of using relatively simple parameters to assess the reflectance of chocolate. The use of the parameter, which is total reflectance, requires the use of relatively advanced equipment, while the brightness analysis - as emphasized by the Reviewer - is a simple and quick measurement.

The figure has been corrected to make it more legible.

In conclusion, the paper has serious scientific problems, does not contribute to the development of a credible method for assessing the ability of packaging to protect food in this case chocolate from radiation and I do not recommend it for publication in a prestigious journal such as Foods.

Reviewer 2 Report

The title of the article is not inadequate to the research part. The tested material is chocolate only but the experiment did not contain a combination of chocolate and packaging (it was not defined), therefore "packaging optimization" did not actually take place in the presented experiment.

Table 1 - lists the tested chocolates, but the analysis of the results does not refer to its variants, such as milk, dessert, and stevia and also does not apply to packaging materials.

Calling chocolate a "functional food" for the purposes of this article is a misuse.

Table 2. illegible reflectance values.

Figure 2. The title in the diagram and description is duplicated. Bar charts lack scale and axis description, making the data difficult to understand.

Figure 3, 4. The title in the diagram and description is duplicated.

The discussion in point 4 is broader and quite interesting, but not clearly connected to the research part.

Author Response

Dear Reviewer,

I am very grateful for the valuable opinions and remarks. I agree with all the comments and I have referred to them as best as possible in a revised version of the manuscript. List of corrections is presented below:

  • The title of the article is not inadequate to the research part. The tested material is chocolate only but the experiment did not contain a combination of chocolate and packaging (it was not defined), therefore "packaging optimization" did not actually take place in the presented experiment.

The title was changed to:

Protection of functional food against solar radiation -  durability of the product and its spectral properties on the example of chocolate

  • Table 1 - lists the tested chocolates, but the analysis of the results does not refer to its variants, such as milk, dessert, and stevia and also does not apply to packaging materials.

Trying to show all the results for each sample or even for each type of chocolate made the number of figure very large. The readability of the results was also very low. At the same time, the range of variability of the examined parameters for all tested chocolate was relatively low, therefore it was decided to present the results in such a way.

  • Calling chocolate a "functional food" for the purposes of this article is a misuse.

Of course, we agree with the Reviewer's opinion that calling chocolate a functional food is a misuse. Nevertheless, among the examined chocolate there are those that meet this condition, e.g. Viminella, Bartfan, which is enriched with vitamin D and vitamin K.

  • Table 2. illegible reflectance values.

Corrected.

  • Figure 2. The title in the diagram and description is duplicated. Bar charts lack scale and axis description, making the data difficult to understand.

Figure 3, 4. The title in the diagram and description is duplicated.

Corrected.

  • The discussion in point 4 is broader and quite interesting, but not clearly connected to the research part.

The discussion was expanded by adding sections that better correspond to the results.

Reviewer 3 Report

The problem of sunlight effect on the product is important and often overlooked. Meanwhile, the degradation of the nutritional value of the product can be caused even by a short exposure to sunlight. Hence, the evaluated work brings important information that can be taken into account in the design of food packaging. In terms of manuscript content, several points need to be clarified:
- Chocolates are packaged in packages that do not let sunlight through. Therefore, the authors should justify why they studied the effect of sunlight on this product.
line 89-90 where the chocolates came from. Was it store-bought or obtained directly from the manufacturer.
table 1. in my opinion the air content in the chocolate may falsify the results so the density of each chocolate sample should be given
figure 1. do the presented tests refer to one kind of chocolate or are they average values
figure 2. you should add confidence interval to the graph and standard deviation to the bars
figure 3 as above
figure 4 as above
discussion section. The results are not well discussed based on the results obtained by other researchers. Hence, I propose to rewrite this section. The discussion section could be divided into three paragraphs. The first paragraph highlights the findings of the study, reviews the results, discusses the results, and makes claims. The second paragraph provides evaluation, analysis, explanation, literature references, and implications. Finally, the third paragraph talks about the study's limitations and recommendations.
Section conclusion
This section needs to be written again. There is a lack of emphasis on the novelty aspects of this research. Also, in this type of study, readers expect some advice in the conclusion: a concise statement of results applicability with confidence interval can be offered.

Author Response

Dear Reviewer,

I am very grateful for the valuable opinions and remarks. I agree with all the comments and I have referred to them as best as possible in a revised version of the manuscript. List of corrections is presented below:

  • The problem of sunlight effect on the product is important and often overlooked. Meanwhile, the degradation of the nutritional value of the product can be caused even by a short exposure to sunlight. Hence, the evaluated work brings important information that can be taken into account in the design of food packaging. In terms of manuscript content, several points need to be clarified:
    - Chocolates are packaged in packages that do not let sunlight through. Therefore, the authors should justify why they studied the effect of sunlight on this product.

The authors' preliminary research indicates that the cardboard chocolate packaging is ineffective and transmits light radiation. At the same time, more and more chocolate have a "window" in their package through which the consumer can see the product. In addition, the proposed method can be used, as mentioned in the manuscript, not only in the field of protection against solar radiation but also against other infrared range - emitted e.g. through shop lighting systems. Infrared penetrates cardboard packaging much better than radiation in the lower wavelength range. Relevant excerpt has been added in the discussion.

  • line 89-90 where the chocolates came from. Was it store-bought or obtained directly from the manufacturer.

The chocolate were bought in the store. Nevertheless, the selection criteria were hat the use-by date was as long as possible.

  • table 1. in my opinion the air content in the chocolate may falsify the results so the density of each chocolate sample should be given

All tested chocolate did not contain any visible air bubbles. In the course of testing chocolate samples (for the purpose of another publication), microtomographic tests were also carried out, which confirmed the negligible air content. In the table, the samples are arranged according to the cocoa content.

figure 1. do the presented tests refer to one kind of chocolate or are they average values

The results presented in figure 1 (figure 2 after corrections) are the average for all tested chocolate. Due to the relatively large number of samples and the small range of reflectance variability depending on the type of chocolate, it was decided to present the dependence of reflectance on the spectral range in figure 1 (figure 2 after corrections). An attempt to present the results for each sample or even each type of chocolate (dessert, milk, dark) made it difficult to visualize the results.

  • figure 2. you should add confidence interval to the graph and standard deviation to the bars
  • figure 3 as above.
    figure 4 as above.

Corrected.

  • discussion section. The results are not well discussed based on the results obtained by other researchers. Hence, I propose to rewrite this section. The discussion section could be divided into three paragraphs. The first paragraph highlights the findings of the study, reviews the results, discusses the results, and makes claims. The second paragraph provides evaluation, analysis, explanation, literature references, and implications. Finally, the third paragraph talks about the study's limitations and recommendations.

Fragments that correspond better with the results have been added. At the same time, an attempt to divide the discussion into sections is not possible due to the limitations of the publishers, which indicate that there should be one paragraph - discussion.

A paragraph on limitations of the study has also been added.

  • Section conclusion
    This section needs to be written again. There is a lack of emphasis on the novelty aspects of this research. Also, in this type of study, readers expect some advice in the conclusion: a concise statement of results applicability with confidence interval can be offered.

Conclusions section has been corrected.

Round 2

Reviewer 1 Report

The authors completed and corrected the manuscript. However, I remain skeptical about the ability of this study to lead to packaging optimizations even if they have clear windows. I mention that even if they are transparent, these materials used for packaging windows have absorption spectra in the ultraviolet or infrared range, which would completely change the spectrum of the solar radiation that reaches the chocolate. Changing the title also changes the approach to the problem and makes it scientifically acceptable.

However, I recommend that the authors do not duplicate the presentation of the results both in the graph and in the table (Fig. 2 Table 1) and to give up the graphs in the form of bars, and to keep only the newly added ones.

Author Response

Prof. PhD. Sławomir Wilczyński                                    Sosnowiec, 18th June 2022

Department of Basic Biomedical Science

Faculty of Pharmaceutical Sciences in Sosnowiec

Medical University of Silesia in Katowice, Poland

Dear Reviewer,

We sincerely thank you for all your comments that have certainly contributed to the higher scientific value of the manuscript. In the revised version of the manuscript, we tried to remove all its flaws listed by the Reviewer.

  • The authors completed and corrected the manuscript. However, I remain skeptical about the ability of this study to lead to packaging optimizations even if they have clear windows. I mention that even if they are transparent, these materials used for packaging windows have absorption spectra in the ultraviolet or infrared range, which would completely change the spectrum of the solar radiation that reaches the chocolate.

The authors are currently conducting research in which they verified the radiation protection properties of packaging for selected food products and drugs. The requirements for medicinal products are even higher than for food (they must meet more stringent standards), and yet the research conducted shows that they are not fully effective in protecting against solar radiation. At this stage, it is difficult to come up with definite conclusions - the research is in progress and will last at least a few more months - nevertheless, it can already be indicated that: plastics protect more or less 30-50% against ultraviolet radiation, and less than 25% against infrared radiation. This is due to the fact that the higher the wavelength (as shown also by the authors in this manuscript), the more effective (deeper) radiation penetration. Hence, it is much more difficult to stop infrared radiation and the packaging can only do this selectively. On the other hand, the use of such a material that selectively absorbs / reflects radiation in a relatively narrow infrared spectrum (the spectrum range which is most strongly absorbed by the product) is technologically possible. At the same time, the research already carried out shows that the most effective method is aluminum foil packaging. This is because metals have free electrons (Fermi-Dirac electron gas) that are not bound to ions forming the crystal lattice. They are electrons from the conduction band, in which there are three types of levels: fully occupied, partially occupied and empty. Free electrons under the influence of incident radiation are subject to variable electric field strength and begin to vibrate with the same frequency as incident radiation (with resonance frequency). The free electron layer on the metal surface absorbs the wave energy, which makes it weaker. The deeper electrons vibrate with a lower amplitude, because the penetrating wave is getting weaker. Therefore, the radiation reaches only a certain depth, extremely small, on the order of 10-10m, and thus even a very thin metal foil, in this case aluminum, is non-transparent for a relatively wide spectral range.

  • Changing the title also changes the approach to the problem and makes it scientifically acceptable.

A new title was proposed: The use of hemispherical directional reflectance to evaluate the interaction of food products with radiation in the solar spectrum.

  • However, I recommend that the authors do not duplicate the presentation of the results both in the graph and in the table (Fig. 2 Table 1) and to give up the graphs in the form of bars, and to keep only the newly added ones.

Corrected. Table 2 that duplicated the results has been deleted. In the manuscript, the graphs in the form of bars were delated, but in the track changes function in MS Word (which is required by the editors), they are still visible.

Reviewer 3 Report

Manuscript in its present form can be accepted for publication.

Author Response

Prof. PhD. Sławomir Wilczyński                                    Sosnowiec, 18th June 2022

Department of Basic Biomedical Science

Faculty of Pharmaceutical Sciences in Sosnowiec

Medical University of Silesia in Katowice, Poland

Dear Reviewer,

We sincerely thank you for the review. Your suggestions certainly contributed to the higher scientific value of the manuscript.
